# On-the-fly learning of adaptive strategies with bandit algorithms

**Rashid Bakirov**                                                RBAKIROV@BOURNEMOUTH.AC.UK
*Department of Computing and Informatics, Bournemouth University, Poole, UK.*

**Damien Fay**
*INFOR/Logicblox, Atlanta, GA, USA*

**Bogdan Gabrys**
*Advanced Analytics Institute, University of Technology Sydney, Ultimo, Australia*

## Abstract

Automation of machine learning model development is increasingly becoming an established research area. While automated model selection and automated data pre-processing have been studied in depth, there is, however, a gap concerning automated model adaptation strategies for streaming data with non-stationarities. This has previously been addressed by heuristic generic adaptation strategies in the batch streaming setting. While showing promising performance, these strategies contain some limitations. In this work, we propose using multi-armed bandit algorithms for learning adaptive strategies from incrementally streaming data on-the-fly. Empirical results using established bandit algorithms show a comparable performance to two common stream learning algorithms.

## 1. Introduction

Machine learning on non-stationary streaming data is a common situation part of which is adapting the model to changes in the underlying data generating process (DGP). While automated model selection and automated data pre-processing have been studied in depth, there is, however, a gap concerning automated model adaptation strategies for streaming data with non-stationarities. In many situations one cannot expect a static model trained on historical data to maintain performance as time proceeds. Here *adaptation* is defined as changes in model *training set*, *parameters* and *structure* all designed to track changes in the underlying DGP over time.

To cope with non-stationary data, many proposed algorithms for machine learning on streaming data involve one or more mechanisms for adapting the model, which will be further referred to as *adaptive mechanisms (AM)*. Deploying various AMs greatly increases the performance of models, however, in most of the cases, the AMs deployment choice (which we will refer to as *adaptation strategy*) is tied to the custom algorithm design choice. With the multitude of available AMs, the design of the adaptation strategy can be a tedious and time consuming task.

Despite advances in automated machine learning, we note that there is a gap concerning automated development of models' adaptation strategy. This has been previously addressed using meta level heuristic adaptation strategies (Bakirov et al., 2021). While showing promising results, these strategies had some shortcomings; they a) required a batch setting, b) increased the runtime and c) did not provide any theoretical performance guarantees. Here, we address these issues by learning the adaptation strategies on-the-fly, this is, *during the data stream process* using the theoretically sound multi-armed bandit algorithms on incremental data. Multi-armed bandit algorithms are designed to maximise a cumulative reward during a sequential process where decisions are made at each step from a limited set of possibilities. This is ideal for solving our problem as adaptation strategies seek to minimise the mean loss over many steps using a small set of AMs. In addition, the theoretical properties of the bandit algorithms provide some optimality guarantees for adaptive strategies.

In this paper, we first formulate the AM selection in the bandit scenario. We then test several prominent bandit algorithms for the Dynamic Weighted Majority (DWM) (Kolter and Maloof, 2007)

and Paired Learner (PL) (Bach and Maloof, 2010) strategies for AM selection. We compare the results to the original DWM algorithm using 26 synthetic and 7 real-world classification data sets. We conclude empirically that the bandit based adaptive strategies leads to comparable performance with the custom strategies. Our contributions are a) a novel formulation of AM selection as a bandit learning problem, and b) an empirical analysis of relevant bandit learning algorithms on AM selection.

## 2. Background

Recently AutoML for streaming non-stationary data has became a focus of the community's attention. An approach to adaptation to changing environments was proposed in (Martín Salvador et al., 2016) where repeated automated deployment of Auto-WEKA for Multi-Component Predictive Systems (MCPS) to learn from new batches of data was used for life-long learning and the adaptation of complex MCPS when applied to changing streaming data from process industries. Celik and Vanschoren (2021) represent a development of this idea with the inclusion of drift detection and the experimentation using several open source AutoML frameworks. An interesting approach closely tied with the Auto_Sklearn is described in (Madrid et al., 2019) where authors propose using the ensemble nature of this framework to deal with streaming data, by adapting the weights of experts and adding new ones. Biedenkapp et al. (2020) report promising results using reinforcement learning for dynamic algorithm configuration in online settings. Wu et al. (2021) introduce a champion-challenger scheme for online AutoML, replacing an existing model with dynamically created new models with bounds on regret. Bakirov et al. (2021) describe the framework of model adaptation with multiple adaptive mechanisms (AMs) and propose heuristics algorithms for automated adaptive strategies based on flexible deployment of these AMs.

## 3. Formulation

To formalise the adaptation with multiple AMS, we use the framework from (Bakirov et al., 2021), adopted for incremental learning. We consider the predictive method at time $t$ as a function $\hat{y}_t = f_t(\boldsymbol{x}_t, \Theta_f)$. where $\hat{y}_t$ is the prediction, $f_t$ is the prediction function, and $\Theta_f$ is the associated parameter set. Our estimate, $f_t$, evolves via adaptation with each $t$-th data instance.

We denote the *a-priori* predictive function at batch $t$ as $f_t^-$, and the *a-posteriori* predictive function, i.e. the adapted function given the observed output, as $f_t^+$. An *adaptive mechanism*, $g(\cdot)$, may thus formally be defined as an operator which generates an updated prediction function based on the instance $\{\boldsymbol{x}_t, y_t\}$ and other optional inputs. This can be written as $g_t(\boldsymbol{x}_t, y_t, \Theta_g, f_t^-, \hat{\boldsymbol{y}}_t) : f_t^- \to f_t^+$ or alternatively as $f_t^+ = f_t^- \circ g_t$ for conciseness. Note $f_t^-$ and $\hat{\boldsymbol{y}}_t$ are optional arguments and $\Theta_g$ is the set of parameters of $g$. The function is propagated onto the next data instance as $f_{t+1}^- = f_t^+$ and predictions themselves are always made using the *a-priori* function $f_t^-$.

We examine a situation when a choice of multiple, different AMs, $\{\emptyset, g_1, ..., g_H\} = G$, is available. Any AM $g_{h_t} \subset G$ can be deployed after each data instance arrival, where $h_t$ denotes the AM deployed for $t$-th data instance. As the history of all adaptations up to the data instance, $t$, have in essence created $f_t^-$, we call that sequence $g_{h_1}, ..., g_{h_t}$ an *adaptation sequence*. Note that we also include the option of applying no adaptation denoted by $\emptyset$, thus all adaptive algorithms having at least the option of *not adapting* fit multiple AMs framework. In this formulation, only one element of $G$ is applied for each data instance. Deploying multiple adaptation mechanisms on the same batch are accounted for with their own symbol in $G$.

Using data instance $\{\boldsymbol{x}_t, y_t\}$ for adaptation, it is possible to obtain $H$ predictive models, $f_t^- \circ g_1, \cdots, f_t^- \circ g_H$. After the true label is revealed, the reward of the selected model in $f_t^- \circ g_{h_t}$, $r_{h_t}$ can be calculated[1]. Thus, the task of coming up with the adaptation strategy amounts to the selection of a predictive model from $F_t$ (equivalent to the selection of $g_{h_t}$ from $G$) for all $t \geq 2$.

---

1. In this work we use $r = 1$ if $y = \hat{y}$ and $r = 0$ otherwise.

It is easy to note that if we assume a fixed unknown reward distribution for each AM, the problem of learning an adaptive strategy is a classical multi-armed bandit problem (Lattimore and Szepesvári, 2020) where the arms are the set $G$ of AMs. Multi-armed bandit algorithms bound *regret*, that is the difference between the chosen and the single best action's reward, sub-linearly depending on the number of rounds $T$ under some conditions. For adaptive strategy learning, a sub-linear regret bound means that the cumulative reward $R_t = \sum_{t=1}^{T} r_t$ using the bandit algorithm for the choice of AMs will converge to this of the optimal single AM selection.

There is a caveat which makes the bounds of classical bandit algorithms not applicable[2] to the AM selection problem. These algorithms assume that the distribution of AMs' rewards are time invariant. This assumption would not hold in stream data scenario as reward distributions will depend on the context such as non-stationarities in data and the choice of previously deployed AMs. This can be handled by the *contextual* bandit algorithms (Tewari and Murphy, 2017), which take additional information such as these into account.

## 4. Experiments

Our experiments are based on the DWM and PL algorithms. DWM is a dynamic ensemble and PL a combination of stable and reactive learners, both very common settings for the stream learning. Both of these settings include multiple AMs, not all of which are used in DWM and PL. For our experiments we will use both settings; a) using *all AMs* and b) as using only the AMs which were part of the original algorithms (*custom AMs*). These options simulate prior information available at the design of adaptive strategy; if known, the poor performing AMs would typically not be included in the set $G$. Without such prior information, it would make sense to use all available AMs for adaptive strategy learning. Used datasets are given in supplementary materials. Below we present the details of stream learning methods and the used bandit algorithms.

### 4.1 Dynamic Weighted Majority style adaptation

DWM is a dynamic weighted classifiers ensemble which enables all three adaptation possibilities common to these methods; the individual retraining or incremental learning of the experts with new data, adjustment of the weights and addition/removal of experts. These can be combined in all possible ways resulting in 8 AMs. We will consider 6 of them as all possible AMs[3].

**AM1** (No adaptation). No changes are applied to the predictive model, corresponding to $\emptyset$.

**AM2** (Incremental Learning). Each predictor is updated with a new data instance.

**AM3** (Weights Update and Experts Pruning). Weights of predictors which misclassify the current data instance are decreased and experts with weights lower than fixed threshold are removed.

**AM4**. AM2 (Weights Update and Experts Pruning) followed by AM1 (Incremental Learning).

**AM5**. AM2 (Weights Update and Experts Pruning) followed by the creation of a new expert.

**AM6**. AM2 (Weights Update and Experts Pruning) followed by AM1 (Incremental Learning) followed by the creation of a new expert.

**DWM original adaptive Strategy.** At time $t$, after an arrival of new instance $\{\boldsymbol{x}_t, y_t\}$, predictors make predictions and the final label is calculated as shown earlier in this section. Then, all predictors learn on this instance and update their weights (AM4) and the final prediction is calculated. If an instance is misclassified, a new predictor with the weight of one is created (AM6).

---

2. In practice the algorithms are still applicable and may provide good results.
3. Excluding the scenarios where experts are added without weights update leading to unlimited expert numbers.

### 4.2 Paired Learner style adaptation

PL maintains two learners - a *stable* learner which is updated with all of incoming data and which is used to make predictions, and a *reactive* learner, which is trained only on a window of the most recent data. For this method, three adaptive mechanisms are possible, which are described below.

**AM1** (No adaptation). No changes are applied to the predictive model, corresponding to $\emptyset$.

**AM2** (Updating stable learner). Stable learner is updated with a new data instance.

**AM3** (Switching to reactive learner). Stable learner is discarded and replaced by reactive learner.

**PL (custom adaptive strategy).** Original PL adaptive strategy revolves around comparing the accuracy values of stable ($u_s^t$) and reactive ($u_r^t$) learners on each batch of data. Every time when $u_s^t < u_r^t$ a change counter is incremented. If the counter is higher than a defined threshold $\theta$, an existing stable learner is discarded and replaced by the reactive learner, while the counter is set to 0. As before, a new reactive learner is continued to be trained with the most recent data.

### 4.3 Bandit algorithms

For learning of adaptive strategies we have experimented with four bandit algorithms, including two variants of LinUCB listed below. For each of these algorithms we consider a cold-start scenario, where we start with uniform estimated rewards per AM and a warm-start (denoted below with (w)), where we run the algorithm on a dataset beforehand twice and use the resulting estimated rewards to report the outcome of the third run [4]. The warm start scenario simulates processing extended data streams as well as reduces the randomness effects in algorithms, all of which use random selection for the cases when two or more AMs have the same estimated reward.

The most basic out of three, $\epsilon$-greedy algorithm (Lattimore and Szepesvári, 2020) estimates the rewards of the actions at time $t$ as mean accumulated reward for this action: $E(r_g) = \frac{R_{g,t}}{t}$. Then, the action with the highest reward is chosen with a probability of $1 - \epsilon$, (so called exploitation) and a random action is chosen with a probability of $\epsilon$ (exploration).

The second bandit algorithm is Kullback-Leibler Upper Confidence Bounds (KL-UCB) (Garivier and Cappé, 2011), a superior variation of the standard Upper Confidence Bounds (UCB) algorithm (Lattimore and Szepesvári, 2020). UCB chooses the action with a highest upper confidence bound of reward at each time step. While time-invariant rewards assumption that KL-UCB has does not hold in our case, meaning that its reward bounds are not applicable, it still provides good results.

To satisfy the condition of rewards' dependency on non-stationarity and previous actions (the context), we include a contextual bandit algorithm LinUCB (linear UCB) (Li et al., 2010). It assumes a linear relationship between the context and the reward, which is estimated via ridge regression. For a standard version of LinUCB, at time $t$ we include two context variables, the result of the last classification (1 if $\hat{y}_{t-1} = y_{t-1}$, 0 otherwise) and the last deployed action $h_{t-1}$. For experiments with sets of all possible AMs both for DWM and PL adaptations styles, we have also used hybrid LinUCB, which also includes the context on the actions. LinUCB has the regret bound of $\tilde{\mathcal{O}}(\sqrt{KdT})$ where $K$ is the number of arms and $d$ is number of context features.

## 5. Results

### 5.1 Dynamic Weighted Majority

**Synthetic data.** The results of the experiments using DWM AMs are shown via Nemenyi plots on Figure 1a,b. With all available AMs, DWM is a leader, however KL-UCB(w) and LINUCB HYBRID(w) achieve statistically comparable performance. LINUCB algorithms show a relatively poor performance. For DWM custom AMs DWM again has the best mean performance rank. This time LINUCB algorithms provide comparable outcome.

---

4. Note that the classifiers are trained from the scratch to avoid data leakage, and while predicting on the instances for which the reward was already calculated may be overly optimistic, the results don't show this bias.

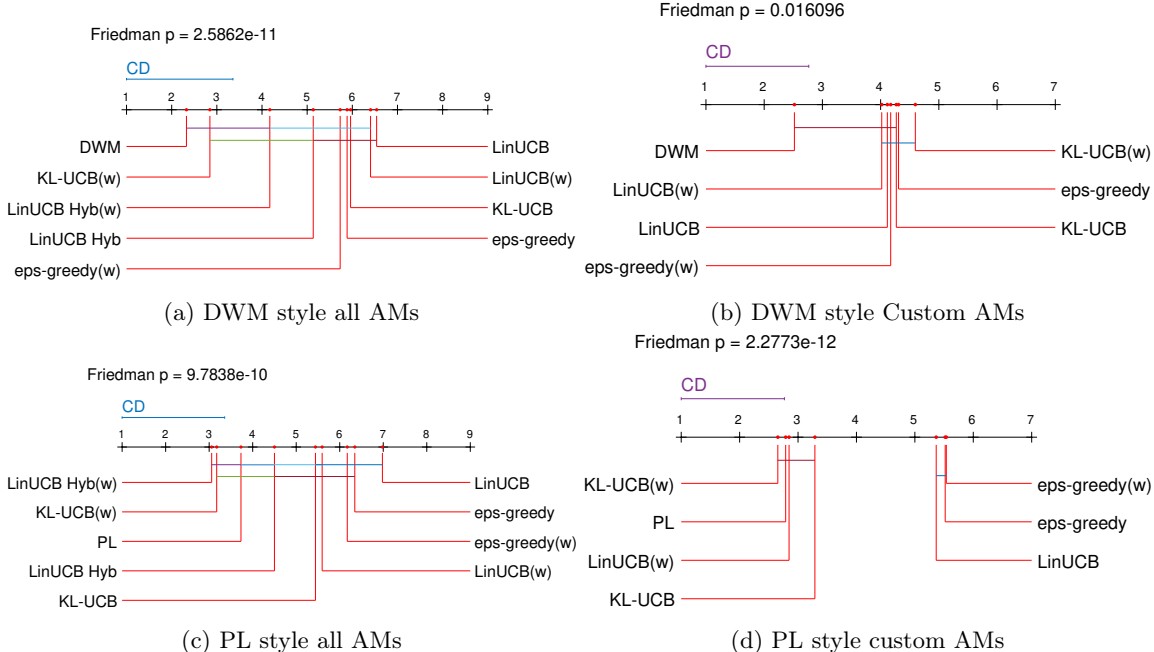

Figure 1: Nemenyi plots (lower is better) of adaptation using bandit-based and custom adaptive strategies on synthetic datasets.

**Real data.** Results on real data on all AMs (Table 1a) show DWM having the top accuracy for most of the datasets. However, only in three cases, the performance of *PL* is more than 1% higher than the runner-up algorithm. Among bandit based methods KL-UCB(w) appears to be a strong contender, while $\epsilon$-GREEDY is lagging behind. The situation is different when considering custom AMs only (Table 2a), as algorithms don't have to test the "weak" AMs (e.g. "do nothing") and suffer a possible reward penalty. Here LINUCB algorithms, especially LINUCB(w) are leaders, with KL-UCB also showing good performance.

### 5.2 Paired Learner

**Synthetic data.** The results of the experiments using PL AMs are shown via Nemenyi plots on Figure 1c,d. With all available AMs many bandit based adaptive strategies achieve comparable average performance to the original PL algorithm, with both LINUCB HYBRID(w) and KL-UCB(w) being slightly better. LINUCB and $\epsilon$-GREEDY based strategies were less successful for this case. Similarly for PL custom AMs KL-UCB(w) shows a slightly better performance.

**Real data.** Results on real data on all AMs (Table 1b) show a slightly different picture, with PL having the top accuracy for most of the datasets. However, the performance of *PL* is more than 1% higher than the runner-up algorithm only in two cases. Among bandit based methods both LinUCB and KL-UCB appear to be strong contenders, while $\epsilon$-GREEDY is lagging behind. Similar situation is observed using custom AMs only (Table 2b). Here KL-UCB is a strong runner up to PL.

## 6. Conclusions

In this paper we have used several bandit algorithms for learning the adaptive strategies for streaming data given a set of adaptive mechanisms on-the-fly. These methods provide easy automation of the adaptation process, avoiding the need to come up with the custom adaptation strategy, some of

Table 1: Accuracy values of adaptation for bandit-based with all AMs and custom adaptive strategies on real datasets. Top accuracy values are bold, runner-ups are bold-italic.

| a) DWM style | LIN UCB | LIN UCB(w) | LinUCB HYB | LinUCB HYB(w) | KL-UCB | KL-UCB(w) | ε-GRD | ε-GRD(w) | DWM |
|---|---|---|---|---|---|---|---|---|---|
| Electricity | 0.855 | 0.843 | 0.853 | 0.853 | 0.845 | *0.872* | 0.847 | 0.847 | **0.874** |
| Power | 0.592 | 0.539 | *0.675* | 0.544 | 0.522 | 0.669 | 0.651 | 0.655 | **0.677** |
| Contraceptive | 0.418 | *0.453* | 0.412 | 0.381 | 0.379 | **0.465** | 0.379 | 0.388 | 0.399 |
| Iris | 0.748 | 0.864 | 0.808 | 0.846 | 0.866 | 0.873 | *0.884* | 0.849 | **0.891** |
| Yeast | 0.403 | 0.468 | 0.278 | *0.468* | 0.468 | **0.469** | 0.312 | 0.314 | 0.297 |
| Gas | 0.893 | 0.916 | 0.852 | 0.889 | 0.893 | *0.925* | 0.919 | 0.919 | **0.943** |
| Gestures | 0.917 | 0.915 | 0.911 | 0.923 | 0.913 | *0.926* | 0.883 | 0.881 | **0.937** |
| **b) PL style** | LIN UCB | LIN UCB(w) | LinUCB HYB | LinUCB HYB(w) | KL-UCB | KL-UCB(w) | ε-GRD | ε-GRD(w) | PL |
| Electricity | 0.860 | *0.862* | 0.861 | 0.859 | 0.859 | 0.858 | 0.839 | 0.840 | **0.866** |
| Power | 0.508 | 0.491 | 0.625 | 0.539 | 0.588 | **0.678** | 0.653 | 0.644 | *0.645* |
| Contraceptive | **0.445** | *0.445* | 0.445 | 0.434 | 0.434 | 0.433 | 0.410 | 0.415 | 0.413 |
| Iris | 0.730 | 0.858 | 0.856 | 0.860 | *0.872* | 0.733 | 0.870 | 0.828 | **0.879** |
| Yeast | 0.284 | 0.385 | 0.289 | *0.470* | **0.470** | 0.337 | 0.288 | 0.287 | 0.320 |
| Gas | 0.816 | 0.901 | *0.910* | 0.891 | 0.897 | 0.898 | 0.887 | 0.889 | **0.929** |
| Gestures | 0.848 | 0.849 | 0.841 | *0.851* | 0.847 | 0.845 | 0.775 | 0.773 | **0.894** |

Table 2: Accuracy values of adaptation for bandit-based with custom AMs and custom adaptive strategies on real datasets. Top accuracy values are bold, runner-ups are bold-italic.

| a) DWM style | LIN UCB | LIN UCB(w) | KL-UCB | KL-UCB(w) | ε-GRD | ε-GRD(w) | DWM |
|---|---|---|---|---|---|---|---|
| Electricity | 0.874 | 0.874 | **0.875** | *0.875* | 0.873 | 0.874 | 0.874 |
| Power | 0.673 | **0.679** | *0.677* | 0.675 | 0.676 | 0.675 | 0.677 |
| Contraceptive | *0.403* | **0.407** | 0.398 | 0.395 | 0.396 | 0.395 | 0.399 |
| Iris | 0.889 | 0.882 | 0.878 | **0.893** | *0.891* | 0.884 | *0.891* |
| Yeast | *0.378* | **0.393** | 0.330 | 0.312 | 0.325 | 0.364 | 0.297 |
| Gas | *0.939* | 0.938 | 0.937 | 0.936 | 0.939 | 0.939 | **0.943** |
| Gestures | *0.930* | *0.930* | 0.928 | 0.929 | 0.929 | 0.929 | **0.937** |
| **b) PL style** | LIN UCB | LIN UCB(w) | KL-UCB | KL-UCB(w) | ε-GRD | ε-GRD(w) | PL |
| Electricity | 0.860 | 0.861 | *0.861* | 0.860 | 0.860 | 0.859 | **0.866** |
| Power | 0.637 | 0.640 | *0.670* | **0.675** | 0.628 | 0.628 | 0.645 |
| Contraceptive | 0.409 | 0.409 | 0.408 | 0.409 | **0.413** | 0.408 | **0.413** |
| Iris | **0.886** | 0.867 | 0.853 | 0.856 | 0.860 | 0.860 | *0.879* |
| Yeast | 0.292 | 0.305 | *0.332* | **0.333** | 0.297 | 0.294 | 0.320 |
| Gas | 0.901 | 0.901 | 0.902 | *0.907* | 0.901 | 0.904 | **0.929** |
| Gestures | 0.849 | 0.842 | *0.850* | *0.850* | 0.843 | 0.846 | **0.894** |

them offering theoretical guarantees in addition. The results show that some bandit algorithms, such as LinUCB and KL-UCB compare favourably and often beat the custom streaming methods, particularly with warm-start. These algorithms are not computationally intensive, with the most time consuming operation being an inverse calculation of $d^2$ dimensional matrix at each step for LinUCB. Further research includes identifying a well-suited set of context features for contextual bandit algorithms, for example considering a window of recent actions instead of only the last one. Another direction is considering a full-information setting, where all of AM would be updated at each time step instead of limited-information bandit setting, where a reward estimate of only the chosen AM is updated. This is likely to result in faster convergence. To improve the convergence transfer-learning type methods can be considered as well, where rewards learned on one dataset can be used for another dataset.

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

## 7. Supplementary material

Table 3: Algorithms hyper-parameters

| Algorithm | Hyper-parameter selection |
|---|---|
| DWM | Weights decay factor $\beta = 0.5$, expert removal weight threshold $\theta = 0.01$ |
| PL | Reactive learner window size $l = 20$, switching threshold $\theta = 1$ |
| LinUCB | $\alpha = 0.5$ |
| $\epsilon$-greedy | $\epsilon = 0.1$ |
| KL-UCB | $c = 0$ |

Table 4: Synthetic classification datasets used in experiments, with $N$ instances and $C$ classes, from (Bakirov and Gabrys, 2013). Column "Drift" specifies number of drifts/changes in data, the percentage of change in the decision boundary and its type. All datasets have 2 input features.

| # | Data type | $N$ | $C$ | Drift | Noise/overlap |
|---|---|---|---|---|---|
| 1 | Hyperplane | 600 | 2 | 2x50% rotation | None |
| 2 | Hyperplane | 600 | 2 | 2x50% rotation | 10% uniform noise |
| 3 | Hyperplane | 600 | 2 | 9x11.11% rotation | None |
| 4 | Hyperplane | 600 | 2 | 9x11.11% rotation | 10% uniform noise |
| 5 | Hyperplane | 640 | 2 | 15x6.67% rotation | None |
| 6 | Hyperplane | 640 | 2 | 15x6.67% rotation | 10% uniform noise |
| 7 | Hyperplane | 1500 | 4 | 2x50% rotation | None |
| 8 | Hyperplane | 1500 | 4 | 2x50% rotation | 10% uniform noise |
| 9 | Gaussian | 1155 | 2 | 4x50% switching | 0-50% overlap |
| 10 | Gaussian | 1155 | 2 | 10x20% switching | 0-50% overlap |
| 11 | Gaussian | 1155 | 2 | 20x10% switching | 0-50% overlap |
| 12 | Gaussian | 2805 | 2 | 4x49.87% passing | 0.21-49.97% overlap |
| 13 | Gaussian | 2805 | 2 | 6x27.34% passing | 0.21-49.97% overlap |
| 14 | Gaussian | 2805 | 2 | 32x9.87% passing | 0.21-49.97% overlap |
| 15 | Gaussian | 945 | 2 | 4x52.05% move | 0.04% overlap |
| 16 | Gaussian | 945 | 2 | 4x52.05% move | 10.39% overlap |
| 17 | Gaussian | 945 | 2 | 8x27.63% move | 0.04% overlap |
| 18 | Gaussian | 945 | 2 | 8x27.63% move | 10.39% overlap |
| 19 | Gaussian | 945 | 2 | 20x11.25% move | 0.04% overlap |
| 20 | Gaussian | 945 | 2 | 20x11.25% move | 10.39% overlap |
| 21 | Gaussian | 1890 | 4 | 4x52.05% move | 0.013% overlap |
| 22 | Gaussian | 1890 | 4 | 4x52.05% move | 10.24% overlap |
| 23 | Gaussian | 1890 | 4 | 8x27.63% move | 0.013% overlap |
| 24 | Gaussian | 1890 | 4 | 8x27.63% move | 10.24% overlap |
| 25 | Gaussian | 1890 | 4 | 20x11.25% move | 0.013% overlap |
| 26 | Gaussian | 1890 | 4 | 20x11.25% move | 10.24% overlap |

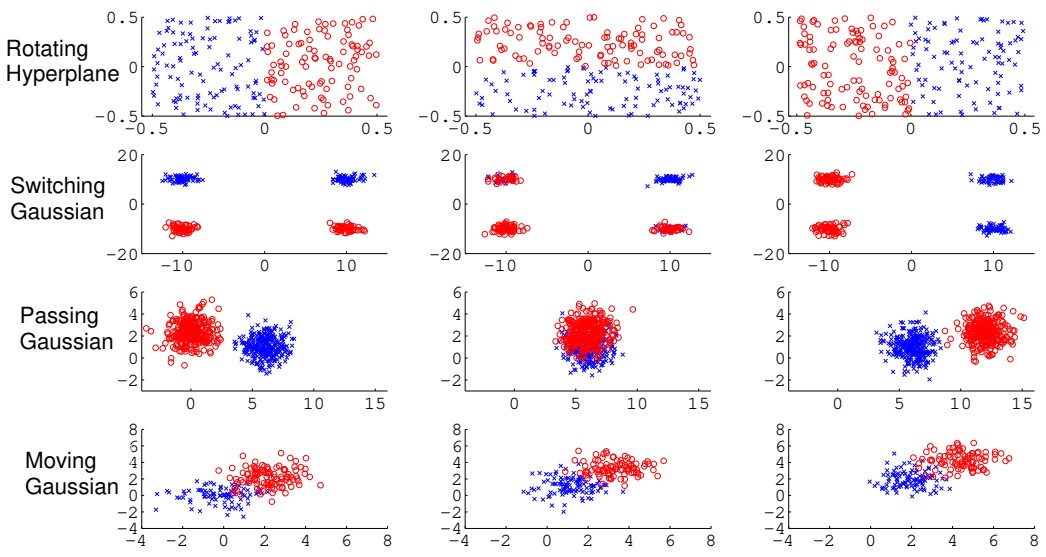

Figure 2: Synthetic datasets visualisation (Bakirov and Gabrys, 2013).

Table 5: Real world classification datasets. $N$ stands for number of instances, $M$ for number of features and $C$ for number of classes.

| # | Name | $N$ | $M$ | $C$ | Brief description |
|---|------|-----|-----|-----|-------------------|
| 1 | Power | 4489 | 2 | 4 | The task is prediction of hour of the day (03:00, 10:00, 17:00 and 21:00) based on supplied and transferred power measured in Italy. (Zhu, 2010; Chen et al., 2015). |
| 2 | Contra-ceptive | 4419 | 9 | 3 | Contraceptive dataset from UCI repository (Newman et al., 1998) with artificially added drift (Minku et al., 2010). |
| 3 | Iris | 450 | 4 | 4 | Iris dataset (Anderson, 1936; Fisher, 1936) with artificially added drift (Minku et al., 2010). |
| 4 | Yeast | 5928 | 8 | 10 | Yeast dataset from UCI repository (Newman et al., 1998) with artificially added drift (Minku et al., 2010). |
| 5 | Gas | 13910 | 129 | 6 | Dataset from chemical sensors utilized in simulations for discrimination task of 6 gases at various levels of concentrations (Vergara et al., 2011). |
| 6 | Gestures | 9873 | 33 | 5 | Dataset composed by features extracted from 7 videos with people gesticulating, for Gesture Phase Segmentation. (Madeo et al., 2013). |
| 7 | Electricity | 27887 | 6 | 2 | Widely used concept drift benchmark dataset thought to have seasonal and other changes as well as noise. Task is the prediction of whether electricity price rises or falls while inputs are days of the week, times of the day and electricity demands (Harries, 1999). |

