# OpenReview forum: "On-the-fly learning of adaptive strategies with bandit algorithms"
_ICML.cc/2021/Workshop/AutoML — AutoML@ICML2021 Poster_

### Official Review · Reviewer_V15U · 2021-06-15
**Dynamic algorithm configuration for online learning**

**Rating:** 6
**Confidence:** 4

**Review:**

# Summary
In this paper, multiple bandit algorithms are investigated to select adaptation actions on-the-fly in an online setting instead of devising a specific hardcoded heuristic or strategy.
The experiments show that the bandit algorithms can compete with well-established approaches to online learning.

# Comments
In the course of the paper the authors attempt to provide a general formalization of the problem, which, from my perspective, lacks a more general AutoML view. Instead the authors rather describe a generic framework for online learning methods that are based on "adaptive mechanisms". Indeed, this is an interesting problem and the considered algorithms seem to yield a good overall performance. From an AutoML perspective the considered methods would still be considered single algorithms. However, I do notice a strong connection to the dynamic algorithm configuration setting:
Biedenkapp, André, et al. "Dynamic algorithm configuration: foundation of a new meta-algorithmic framework." Proceedings of the Twenty-fourth European Conference on Artificial Intelligence (ECAI’20)(Jun 2020). 2020.

The main contribution of the paper consists in the empirical evaluation of the bandit algorithms and comparing each to the baseline algorithms. However, important details for the evaluation are missing, such as the computation time, how performances are determined, and details on the hardware specifications are missing. Code and data are not provided with the paper which could further help to rerun the experiments and understand the experimental setup. This brings me to the next issue. Throughout the paper the authors speak about maximizing cumulative rewards. In Section 4 suddenly regret comes up, but in the result tables accuracy is reported, which is quite confusing. Although requiring notions of bandit algorithms to be known, the evaluation of the algorithms is rather unusual for the bandit community. Here, one would rather expect cumulative regret/reward curves. Speaking about cumulative regret/reward: In footnote 1 the authors state the a correct prediction will receive no reward and a reward of 1 otherwise, which is probably a mistake.

Assuming that the rewards are received properly, I wonder how the Bandit algorithm will learn anything useful at all. Frankly speaking, the results are somewhat surprising to me, since except for LinUCB no context is considered when deciding for an arm. Can you please elaborate on details what strategies the Bandit algorithms follow? Is it a rather consistent strategy, such as update the model every time?

- The definition of g_t looks like a mixture of a function signature and a specific mapping.

- "Otherwise, it would make sense to use all available AMs for adaptive strategy learning.": What is the statement here?

- The warmstarting strategy looks to me like data snooping. Why should one be allowed to already conduct two runs on the same data? Moreover, why consists warmstarting of running the algorithm exactly twice? Why is it legal to already sneak into the data?

- "the last deployed action h_{t-1}" h is not introduced, what is it? What is the motivation for these two features? Only the last decision? What about a larger window to deal with noise and not "overreact" on wrong predictions?

- What are best mean performance ranks?

All in all, I think that more space should be allocated to a clean and self-contained definition of the setting and explanation of design decisions etc instead of pasting Nemenyi plots and two large tables. Both result presentation formats hide a lot of important information regarding online algorithms and do not necessarily provide the desired details to learn something out of the results. The surrounding text is also only a shallow description of what can be seen from the tables. Moreover, the tables are rather hard to read if one wants to check which bandit algorithm performs superior compared to the baseline.

Please double-check the references. For example, the reference Bach and Maloof (2010) looks quite odd.

Concerning references, here is another reference which might be of interest to the authors:
Qingyun Wu and Chi Wang and John Langford and Paul Mineiro and Marco Rossi: ChaCha for Online AutoML. 2021 arXiv abs/2106.04815

# Presentation
The paper would clearly benefit of proofreading. Just a few examples:
"these strategies contain some limitations." => these strategies have some limitations.
"automated development of models' adaptation strategy." => What is this supposed to mean?
"Auto_Sklearn" => auto-sklearn or Auto-sklearn
"Bakirov et al. (2021) describe the framework [...] and proposes heuristics algorithms for automated adaptive strategies" => Bakirov et al. (2021) describe the framework [...] and propose heuristic algorithms for automated adaptive strategies
"the a priori predictive function" => the a-priori predictive function (same for a posteriori)
"thus any adaptive algorithm fits multiple AMs framework" => What is this supposed to mean? I dont understand this sentence.
"the problem of learning of adaptive strategy" => the problem of learning an adaptive strategy
"which were part of the the original algorithms" => double "the"

- AM references in 4.1 are confused. AM2 has the meaning of AM3 in parentheses and AM 1 originally refers to No adaptation instead of Incremental Learning.

---

### Official Review · Reviewer_ooWR · 2021-06-15
**A very interesting idea, but the results are described more optimistic that the figures/tables warrant**

**Rating:** 5
**Confidence:** 4

**Review:**

The authors propose a bandit-based framework for AutoML on data streams. I personally think that this is a very promising idea, however, the results are not extremely convincing yet, and more research needs to be done before this can be communicated to other researchers by means of a workshop publication.

Firstly, I am surprised that some relevant related work is missing. In recent literature, a similar approach is often abstracted as an ensemble, where based on recent performances certain ensemble members get a higher preference. The authors should compare against such techniques, or at least discuss why the comparison is missing.
- Having a Blast: Meta-Learning and Heterogeneous Ensembles for Data Streams, 2015
- Arbitrated Ensemble for Time Series Forecasting, 2017
- The online performance estimation framework: heterogeneous ensemble learning for data streams, 2018
Additionally, it is not clear how the work relates to the work provided by Celik and Vanschoren (2020). Finally, the reference section contains only 5 references coming from the past 5 years (2016 or more recent), making me doubt the knowledge of the authors of the data stream literature a bit. In the references, also some missing entrees are not filled in yet (see Bach and Maloof, 2010)

The results are presented along three dimensions:
- real data vs. generated data
- integrated into DWM vs. integrated into PL
- custom AM's vs all AM's

This results in six sets of experiments. The baselines are not extremely ambitious, with DMW coming from 2007, and PL coming from 2010. As mentioned before, more successful approaches have been developed in the meantime. In only the generated data integrated into PL, one of the various suggested approaches can be claimed superior over the presented baselines. In all other cases, it seems that the baselines are superior or at least competitive to the various approaches.
I respect the authors for showing all the results nonetheless, which gives a clear idea of where the approach can be extended. Clearly, using the custom AM's helps the performance in a positive way.
However, the way that the results are discussed in the introduction and conclusion are not adequate compared to what the tables and plots reveal.

Finally, I would expect a bit more justification of the selected hyperparameters (Table 3), especially given the target audience.

Altogether, I feel like the work would be interesting to discuss. I would strongly encourage the authors to keep working on this topic, as there is a clear need for better AutoML for data streams. However, in its current form, the work is not ready for publication yet.

---

### Decision · Program_Chairs · 2021-06-21

Accept (Poster)